# Keep Doing What Worked:
# Behavior Modelling Priors for Offline Reinforcement Learning

Noah Y. Siegel, Jost Tobias Springenberg, Felix Berkenkamp, Abbas Abdolmaleki, Michael Neunert, Thomas Lampe, Roland Hafner, Nicolas Heess, Martin Riedmiller

DeepMind
{siegeln}@google.com

## Abstract

Off-policy reinforcement learning algorithms promise to be applicable in settings where only a fixed data-set (batch) of environment interactions is available and no new experience can be acquired. This property makes these algorithms appealing for real world problems such as robot control. In practice, however, standard off-policy algorithms fail in the batch setting for continuous control. In this paper, we propose a simple solution to this problem. It admits the use of data generated by arbitrary behavior policies and uses a learned prior – the advantage-weighted behavior model (ABM) – to bias the RL policy towards actions that have previously been executed and are likely to be successful on the new task. Our method can be seen as an extension of recent work on batch-RL that enables stable learning from conflicting data-sources. We find improvements on competitive baselines in a variety of RL tasks – including standard continuous control benchmarks and multi-task learning for simulated and real-world robots. Videos are available at https://sites.google.com/view/behavior-modelling-priors.

## 1 Introduction

Batch reinforcement learning (RL) (Ernst et al., 2005; Lange et al., 2011) is the problem of learning a policy from a fixed, previously recorded, dataset without the opportunity to collect new data through interaction with the environment. This is in contrast to the typical RL setting which alternates between policy improvement and environment interaction (to acquire data for policy evaluation). In many real world domains collecting new data is laborious and costly, both in terms of experimentation time and hardware availability but also in terms of the human labour involved in supervising experiments. This is especially evident in robotics applications (see e.g. Riedmiller et al. 2018; Haarnoja et al. 2018b; Kalashnikov et al. 2018 for recent examples learning on robots). In these settings where gathering new data is expensive compared to the cost of learning, batch RL promises to be a powerful solution.

There exist a wide class of off-policy algorithms for reinforcement learning designed to handle data generated by a behavior policy $\mu$ which might differ from $\pi$, the policy that we are interested in learning (see e.g. Sutton & Barto (2018) for an introduction). One might thus expect solving batch RL to be a straightforward application of these algorithms. Surprisingly, for batch RL in continuous control domains, however, Fujimoto et al. (2018) found that policies obtained via the naïve application of off-policy methods perform dramatically worse than the policy that was used to generate the data. This result highlights the key challenge in batch RL: we need to exhaustively exploit the information that is in the data but avoid drawing conclusions for which there is no evidence (i.e. we need to avoid over-valuing state-action sequences not present in the training data).

As we will show in this paper, the problems with existing methods in the batch learning setting are further exacerbated when the provided data contains behavioral trajectories from different policies $\mu_1, \ldots, \mu_N$ which solve different tasks, or the same task in different ways (and thus potentially execute conflicting actions) that are not necessarily aligned with the target task that $\pi$ should accomplish. We empirically show that previously suggested adaptations for off-policy learning

(Fujimoto et al., 2018; Kumar et al., 2019) can be led astray by behavioral patterns in the data that are consistent (i.e. policies that try to accomplish a different task or a subset of the goals for the target task) but not relevant for the task at hand. This situation is more damaging than learning from noisy or random data where the behavior policy is sub-optimal but is not predictable, i.e. the randomness is not a correlated signal that will be picked up by the learning algorithm.

We propose to solve this problem by restricting our solutions to 'stay close to the relevant data'. This is done by: 1) learning a prior that gives information about which candidate policies are potentially supported by the data (while ensuring that the prior focuses on relevant trajectories), 2) enforcing the policy improvement step to stay close to the learned prior policy. We propose a policy iteration algorithm in which the prior is learned to form an advantage-weighted model of the behavior data. This prior biases the RL policy towards previously experienced actions that also have a high chance of being successful in the current task. Our method enables stable learning from conflicting data sources and we show improvements on competitive baselines in a variety of RL tasks – including standard continuous control benchmarks and multi-task learning for simulated and real-world robots. We also find that utilizing an appropriate prior is sufficient to stabilize learning; demonstrating that the policy evaluation step is implicitly stabilized when a policy iteration algorithm is used – as long as care is taken to faithfully evaluate the value function within temporal difference calculations. This results in a simpler algorithm than in previous work (Fujimoto et al., 2018; Kumar et al., 2019).

## 2 BACKGROUND AND NOTATION

In the following we consider the problem of reinforcement learning, modeling the environment as a markov decision process (MDP) consisting of the continuous states $s \in S$, actions $a \in \mathcal{A}$, and transition probability distribution $p(s_{t+1}|s_t, a_t)$ – describing the evolution of the system dynamics over time (e.g. probability of reaching $s_{t+1}$ from state $s_t$ when executing action $a_t$) – together with the state-visitation distribution $p(s)$. The goal of reinforcement learning is to find a policy $\pi(a|s)$ that maximizes the cumulative discounted return $J(\pi) = \mathbb{E}_\pi[\sum_{t=1}^\infty \gamma^t r(s_t, a_t)|a_t \sim \pi(\cdot|s_t), s_{t+1} \sim p(\cdot|s_t, a_t), s_1 \sim p(\cdot)]$, for the reward function $r(s, a) \in \mathbb{R}$. We also define the state-action value function for taking action $a_t$ in state $s_t$, and thereafter following $\pi$: $Q^\pi(s_t, a_t) = \mathbb{E}_\pi[\sum_{i=t}^\infty \gamma^i r(s_i, a_i)|a_i \sim \pi, s_{i+1} \sim p(\cdot|s_i, a_i)]$, which we can relate to the objective $J$ via $J(\pi) = \mathbb{E}_{s \sim p(\cdot)}[\mathbb{E}_{a \sim \pi(\cdot|s)}[Q^{\pi^*}(s, a)]]$, where $\pi^*$ is the optimal policy. We parameterize the policy $\pi_\theta(a|s)$ by $\theta$ but we will omit this dependency where unambiguous. In some of the experiments we will also consider a setting where we learn about multiple tasks $k \in \{1, \ldots K\}$, each with their own reward function $r_k(s, a)$. We condition the policy and Q-function on the task index $k$ (i.e. $Q_k^\pi$ and $\pi(a|s, k)$), changing the objective to maximize the sum of returns across all tasks.

For the batch RL setting we assume that we are given a dataset $\mathcal{D}_\mu$ containing trajectory snippets (i.e. sub-trajectories of length $N$) $\tau = \{(s_0, a_0), \cdots, (s_T, a_T)\}$, with $\tau \in \mathcal{D}_\mu$. We assume access to the reward function $r$ for the task of interest and can evaluate it for all transitions in $\mathcal{D}_\mu$ (for example, $r$ may be some function of $s$). We further assume $\mathcal{D}_\mu$ was filled, prior to training, by following a set of arbitrary $N$ behavior policies $\mu_1, \cdots, \mu_N$. Note that these behavior policies may try to accomplish the task we are interested in; or might indeed generate trajectories unrelated to the task at hand.

## 3 A LEARNED PRIOR FOR OFFLINE OFF-POLICY RL FROM IMPERFECT DATA

To stabilize off-policy RL from batch data, we want to restrict the learned policy to those parts of the state-action space supported by the batch. In practice this means that we need to approximately restrict the policy to the support of the empirical state-conditional action distribution. This prevents the policy from taking actions for which the Q-function cannot be trained and for which it might thus give erroneous, overly optimistic values (Fujimoto et al., 2018; Kumar et al., 2019). In this paper we achieve this by adopting a policy iteration procedure – in which the policy is constrained in the improvement step. As in standard policy iteration (Sutton & Barto, 2018), the procedure consists of two alternating steps. First, starting with a given policy $\pi_i = \pi_{\theta_i}$ in iteration $i$ (with $\pi_{\theta_0}$ corresponding to a randomly initialized policy distribution), we find an approximate action-value function $Q^{\pi_i}(s, a) \approx \hat{Q}(s, a; \phi_i)$, with parameters $\phi$ (Section 3.1) (as with the policy we will drop the dependence on $\phi_i$ and write $\hat{Q}^{\pi_i}(s, a)$ where unambiguous). Second, we optimize for $\pi_{i+1}$ with respect to $\hat{Q}^{\pi_i}$ subject to a constraint that ensures closeness to the empirical state-conditional action

distribution of the batch (Section 3.2). Iterating these steps, overall, optimizes $J(\pi)$. We realize both policy evaluation and improvement via a fixed number of gradient descent steps – holding $\pi$ and $\hat{Q}^{\pi_i}$ fixed via the use of target networks (Mnih et al., 2015). We refer to Algorithm 1 for details.

## 3.1 POLICY EVALUATION

To learn the task action-value function in each iteration we minimize the squared temporal difference error for a given reward – note that when performing offline RL from a batch of data *the reward $r$ might be computed post-hoc and does not necessarily correspond to the reward optimized by the behavior policies $\mu_1, \cdots, \mu_N$.* The result after iteration $i$ is given as

$$
\phi_i = \arg\min_{\phi} \ \mathbb{E}_{\tau \sim \mathcal{D}_\mu} \Big[ \big( r(s_t, a_t) + \gamma \hat{V}^{\pi_i}(s_{t+1}) - \hat{Q}(s_t, a_t; \phi) \big)^2 \big| (s_t, a_t, s_{t+1}) \sim \tau \Big],
$$
$$
\text{with } \hat{V}^{\pi_i}(s) = \mathbb{E}_{a \sim \pi_i(\cdot|s)} \big[ \hat{Q}(s, a; \phi_{i-1}) \big]
$$
(1)

We approximate the expectation required to calculate $\hat{V}^{\pi_i}(s)$ with $M$ samples from $\pi_i$, i.e. $\hat{V}^{\pi_i}(s) \approx \frac{1}{M} [\sum_{j=1}^{M} \hat{Q}(s, a_j; \phi_{i-1}) \mid a_j \sim \pi_i(\cdot|s)]$. As further discussed in the related work Section 4, the use of policy evaluation is different from the Q-learning approach pursued in Fujimoto et al. (2018); Kumar et al. (2019), which requires a maximum over actions and may be more susceptible to overestimation of Q-values. We find that when enough samples are taken (we use $M = 20$) and the policy is appropriately regularized (see Section 3.2) learning is stable without additional modifications.

## 3.2 PRIOR LEARNING AND POLICY IMPROVEMENT

In the policy improvement step we solve the following constrained optimization problem

$$
\pi_{i+1} = \arg\max_{\pi} \ \mathbb{E}_{\tau \sim \mathcal{D}_\mu} \Big[ \mathbb{E}_{a \sim \pi(\cdot|s)} \big[ \hat{Q}^{\pi_i}(s, a) \big] \big| s \sim \tau \Big]
$$
$$
\text{s.t.} \ \mathbb{E}_{\tau \sim \mathcal{D}_\mu} \big[ \text{KL}[\pi(\cdot|s) \| \pi_{\text{prior}}(\cdot|s)] | s \sim \tau \big] \le \epsilon,
$$
(2)

where $\mathcal{D}_\mu$ is the behavior data, $\pi$ the policy being learned, and $\pi_{\text{prior}}$ is the prior policy. This is similar to the policy improvement step in Abdolmaleki et al. (2018) but instead of enforcing closeness to the previous policy here the constraint is with respect to a separately learned "prior" policy, the *behavior model*. The role of $\pi_{\text{prior}}$ in Equation 2 is to keep the policy close to the regime of the actions found in $\mathcal{D}_\mu$. We consider two different ways to express this idea by learning a prior alongside the policy optimization.

For learning the prior, we first consider simply modeling the raw behavior data. This is similar to the approach of BCQ and BEAR-QL (Fujimoto et al., 2018; Kumar et al., 2019), but we use a parametric behavior model and measure distance by KL; we refer to the related work for a discussion. The behavior model can be learned by maximizing the log likelihood of the observed data

$$
\theta_{\text{bm}} = \arg\max_{\theta_{\text{bm}}} \ \mathbb{E}_{\tau \sim \mathcal{D}_\mu} \left[ \sum_{t=1}^{|\tau|} \log \pi_{\theta_{\text{bm}}}(a_t|s_t) \right],
$$
(3)

where $\theta_{\text{bm}}$ are the parameters of the behavior model prior.

Regularizing towards the behavior model can help to prevent the use of unobserved actions, but it may also prevent the policy from improving over the behavior in $\mathcal{D}_\mu$. In effect, the simple behavior prior in Equation 3 regularizes the new policy towards the empirical state-conditional action distribution in $\mathcal{D}_\mu$. This may be acceptable for datasets dominated by successful trajectories for the task of interest or when the unsuccessful trajectories are not predictable (i.e. they correspond to random behaviour). However, we here are interested in the case where $\mathcal{D}_\mu$ is collected from imperfect data and from multiple tasks. In this case, $\mathcal{D}_\mu$ will contain a diverse set of trajectories – both (partially) successful and actively harmful for the target task. With this in mind, we consider a second learned prior, the advantage-weighted behavior model, $\pi_{\text{abm}}$, with which we can bias the RL policy to choose actions that are both supported by $\mathcal{D}_\mu$ and also good for the current task (i.e. keep doing actions that work).

We can formulate this as maximizing the following objective:

$$\theta_{\mathrm{abm}} = \arg\max_{\theta_{\mathrm{abm}}} \mathbb{E}_{\tau \sim \mathcal{D}_\mu} \left[ \sum_{t=1}^{|\tau|} \log \pi_{\theta_{\mathrm{abm}}}(a_t|s_t) f\left( R\left(\tau_{t:N}\right) - \hat{V}^{\pi_i}(s) \right) \right],$$

$$\text{with } R\left(\tau_{t:N}\right) = \gamma^{N-t}\hat{V}^{\pi_i}(s_N) + \sum_{j=t}^{N-1} \gamma^{j-t} r(s_j, a_j), \tag{4}$$

where $f$ is an increasing, non-negative function, and the difference $R\left(\tau_{t:N}\right) - \hat{V}^{\pi_i}$ is akin to an n-step advantage function, but here calculated off-policy representing the "advantage" of the behavior snippet over the policy $\pi_i$. This objective still tries to maximize the log likelihood of observed actions, and avoids taking actions not supported by data. However, by "advantage weighting" we focus the model on "good" actions while ignoring poor actions. We let $f = 1_+$ (the unit step function with $f(x) = 1$ for $x \geq 0$ and 0 otherwise) both for simplicty – to keep the number of hyperparameters to a minimum while keeping the prior broad – and because it has an intuitive interpretation: such a prior will start by covering the full data and, over time, filter out trajectories that would lead to worse performance than the current policy, until it eventually converges to the best trajectory snippets contained in the data. We note that Equation 4 is similar to a policy gradient, though samples here stem from the buffer and it will thus not necessarily converge to the optimal policy in itself; $\pi_{\theta_{\mathrm{abm}}}$ will instead only cover the best trajectories in the data due to no importance weighting being performed for off-policy data. This bias is in fact desirable in a batch-RL setting; we want a broad prior that only considers actions present in the data. We also note that we tried several different functions for $f$ including exponentiation, e.g. $f(x) = \exp(x)$, but found that choice of function did not make a significant difference in our experiments.

Using either $\pi_{\theta_{\mathrm{bm}}}$ or $\pi_{\theta_{\mathrm{abm}}}$ as $\pi_{\mathrm{prior}}$, Equation 2 can be solved with a variety of optimization schemes. We experimented with an EM-style optimization following the derivations for the MPO algorithm (Abdolmaleki et al., 2018), as well as directly using the stochastic value gradient of $\hat{Q}^{\pi_i}$ wrt. policy parameters (Heess et al., 2015).

It should be noted that, if the prior itself is already good enough to solve the task we can learn $Q^{\pi_{\mathrm{prior}}}$ – e.g. if the data stems from an expert or has sufficiently high quality. In this case learning both $Q^{\pi_{\mathrm{prior}}}$ and $\pi_{\mathrm{prior}}$ becomes independent of the RL policy improvement step; if we then set $\epsilon = 0$, skipping the policy improvement step, we obtain a further simplified algorithm consisting only of learning $\pi_{\theta_{\mathrm{abm}}}$ and $Q^{\pi_{\theta_{\mathrm{abm}}}}$ (see Figure 9 for an ablation).

**EM-style optimization** We can optimize the objective from Equation 2 using a two-step procedure. Following (Abdolmaleki et al., 2018), we first notice that the optimal $\pi$ for Equation 2 can be expressed as $\hat{\pi}(a|s) \propto \pi_{\mathrm{prior}}(a|s) \exp(\hat{Q}^{\pi_i}(s,a)/\eta)$, where $\eta$ is a temperature that depends on the $\epsilon$ used for the KL constraint and can be found automatically by a convex optimization (Appendix B.1). Conveniently, we can sample from this distribution by querying $\hat{Q}^{\pi_i}$ using samples from $\pi_{\mathrm{prior}}$. These samples can then be used to learn the parametric policy by minimizing the divergence $KL(\hat{\pi}\|\pi_{\theta_{i+1}})$, which is equivalent to maximizing the weighted log likelihood

$$\theta_{i+1} = \arg\max_\theta \mathbb{E}_{\tau \sim \mathcal{D}_\mu} \left[ \mathbb{E}_{a \sim \pi_{\mathrm{prior}}(\cdot|s)} \left[ \exp(\hat{Q}^{\pi_i}(s,a)/\eta) \log \pi_\theta(a|s) | s \sim \tau \right] \right], \tag{5}$$

which we optimize via gradient descent subject to an additional trust-region constraint on $\pi_\theta$ given as $KL(\pi_{\theta_i}\|\pi_\theta) < \epsilon_{\mathrm{trust}}$ to ensure conservative updates (Appendix B.1).

**Stochastic value gradient optimization** Alternatively, we can use Langrangian relaxation to turn Equation 2 into an objective amenable to gradient descent. Inserting $\pi_\theta$ for $\pi$ and relaxing results in

$$(\theta_{i+1}, \eta) = \arg\max_\theta \min_\eta \mathbb{E}_{\tau \sim \mathcal{D}_\mu} \left[ \mathbb{E}_{a \sim \pi_\theta(\cdot|s)} \left[ \hat{Q}^{\pi_i}(s,a) \right] + \eta(\epsilon - \mathrm{KL}[\pi_\theta(\cdot|s)\|\pi_{\mathrm{prior}}(\cdot|s)]) \right], \tag{6}$$

for $\eta > 0$ and which we can optimize by alternating gradient descent steps on $\theta$ and $\eta$ respectively, taking the stochastic gradient of the Q-value (Heess et al., 2015) through the sampling of $a \sim \pi_\theta(\cdot, s)$ via re-parameterization. See Appendix B.2 for a derivation of this gradient.

Full pseudocode for our approach can be found in Appendix A, Algorithm 1.

## 4 RELATED WORK

There exist a number of off-policy RL algorithms that have been developed since the inception of the RL paradigm (see e.g. Sutton & Barto (2018) for an overview). Most relevant for our work, some of these have been studied in combination with function approximators (for estimating value functions and policies) with an eye on convergence properties in the batch RL setting. In particular, several papers have theoretically analyzed the accumulation of bootstrapping errors in approximate dynamic programming (Bertsekas & Tsitsiklis, 1996; Munos, 2005) and approximate policy iteration (Farahmand et al., 2010; Scherrer et al., 2015); for the latter of which there exist well known algorithms that are stable at least with linear function approximation (see e.g. Lagoudakis & Parr (2003)). Work on RL with non-linear function approximators has mainly considered the "online" or "growing batch" settings, where additional exploration data is collected (Ernst et al., 2005; Riedmiller, 2005; Ormoneit & Sen, 2002); though some success for batch RL in discrete domains has been reported (Agarwal et al., 2019). For continuous action domains, however, off-policy algorithms that are commonly used with powerful function approximators fail in the fixed batch setting.

Prior work has identified the cause of these failures as extrapolation or bootstrapping errors (Fujimoto et al., 2018; Kumar et al., 2019) which occur due to a failure to accurately estimate Q-values, especially for state-action pairs not present in the fixed data set. Greedy exploitation of such misleading Q-values (e.g. due to a $\max$ operation) can then cause further propagation of such errors in the Bellman backup, and to inappropriate action choices during policy execution (leading to suboptimal behavior). In non-batch settings, new data gathered during exploration allows for the Q-function to be corrected. In the batch setting, however, this feedback loop is broken, and correction never occurs.

To mitigate these problems, previous algorithms based on Q-learning identified two potential solutions: 1) correcting for overly optimistic Q-values in the Bellman update, and 2) restricting the policy from taking actions unlikely to occur in the data. To address 1) prior work uses a Bellman backup operator in which the $\max$ operation is replaced by a generative model of actions (Fujimoto et al., 2018) which a learned policy is only allowed to minimally perturb; or via a maximum over actions sampled from a policy which is constrained to stay close to the data (Kumar et al., 2019) (implemented through a constraint on the distance to a model of the empirical data, measured either in terms of maximum mean discrepancy or relative entropy). To further penalize uncertainty in the Q-values this can be combined with Clipped Double-Q learning (Fujimoto et al., 2018) or an ensemble of Q-networks (Kumar et al., 2019). To address 2) prior work uses a similarly constrained $\max$ also during execution, by considering only actions sampled from the perturbed generative model (Fujimoto et al., 2018) or the constrained policy (Kumar et al., 2019), and choosing the best among them.

Our work is based on a policy iteration scheme instead of Q-learning – exchanging the $\max$ for an expectation. Thus we directly learn a parametric policy that we also use for execution. We estimate the Q-function as part of the policy evaluation step with standard TD-0 backups. We find that for an appropriately constrained policy no special treatment of the backup operator is necessary, and that it is sufficient to simply use an adequate number of samples to approximate the expectation when estimating $V$ (see Equation 1). The only modification required is in the policy improvement step where we constrain the policy to remain close to the adaptive prior in Equation 2. As we demonstrate in the empirical evaluation it is the particular nature of the adaptive prior – which can adapt to the task at hand (see Equation 4) – that makes this constraint work well. Additional measures to account for uncertainty in the Q values could also be integrated into our policy evaluation step but we did not find it to be necessary for this work; we thus forego this in favor of our simpler procedure.

Our policy iteration scheme also bears similarity to previous works that use (relative) entropy regularized policy updates. These use constraints with respect to either a fixed (e.g. uniform) policy (e.g. Haarnoja et al., 2018a) or, in a trust-region like scheme, to the previous policy (e.g. Abdolmaleki et al., 2018). Other work has also focused on policy priors that are optimized to be different from the actual policy but so far mainly in the multi-task or transfer-learning setting (Teh et al., 2017; Galashov et al., 2019; Tirumala et al., 2019; Jaques et al., 2017). The constrained updates are also related to trust-region optimization in action space, e.g. in TRPO / PPO (Schulman et al., 2015; 2017) and MPO (Abdolmaleki et al., 2018), which ensures stable learning by enforcing conservative updates; an idea that can be traced back to Kakade & Langford (2002). Here we take a slightly different perspective: we enforce a trust region constraint not on the last policy in the policy optimization loop (conservative updates) but wrt. the advantage weighted behavior distribution.

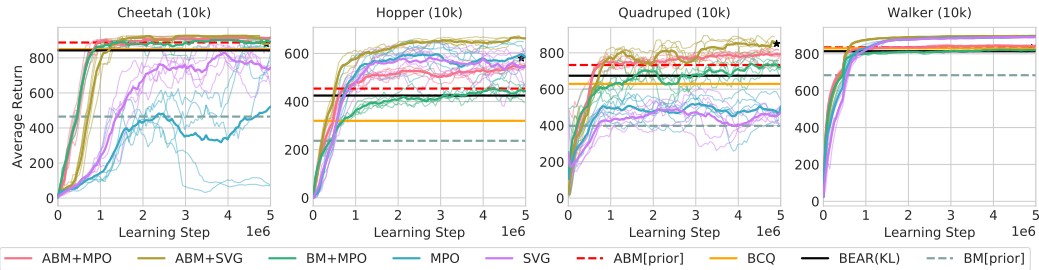

Figure 1: Learning control suite tasks from fixed data. Standard off-policy RL fails on Cheetah, Hopper, Quadruped. Methods that regularize to the data distribution succeed; for hopper the behaviour distribution is multimodal and a standard behavior modlling prior is broad (see BM[prior]) here only ABM succeeds. Best trajectory in data marked with a star.

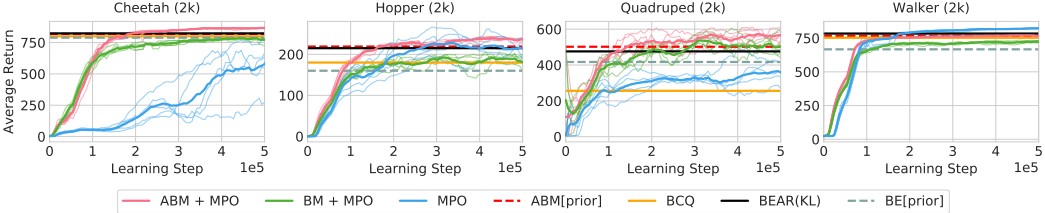

Figure 2: Control suite when using only the first 2k episodes of low-quality data from each run; return for best behavior episode in the data was $718/159/435/728$ (left to right). While plain RL is able to learn on walker, learned priors improve performance and stabilize learning for cheetah and quadruped (with overall lower performance than when learning from good data).

## 5  EXPERIMENTS

We experiment with continuous control tasks in two different settings. In a first set of experiments we compare our algorithm to strong off-policy baselines on tasks from the DeepMind control suite (Tassa et al., 2018) – to give a reference point as to how our algorithm performs on common benchmarks. We then turn to the more challenging setting of learning multiple tasks involving manipulation of blocks using a robot arm in simulation. These span tasks from reaching toward a block to stacking one block on top of another. Finally, we experiment with analogous tasks on a real robot.

We use the same networks for all algorithms that we compare, optimize parameters using Adam (Kingma & Ba, 2015), and utilize proprioceptive features (e.g. joint positions / velocities) together with task relevant information (mujoco state for the control suite, and position/velocity estimates of the blocks for the manipulation tasks). All algorithms were implemented in the same framework, including our reproduction of BCQ and BEAR, and differ only in their update rules. Note that for BEAR we use a KL instead of the MMD as we found this to work well, see appendix. In the multi-task setting (Section 5.1) we learn a task conditional policy $\pi_\theta(a|s, k)$ and Q-function $\hat{Q}_\phi^\pi(s, a, k)$ where $k$ is a one-hot encoding of the task identifier, that is provided as an additional network input. We refer to the appendix for additional details.

### 5.1  CONTROL SUITE EXPERIMENTS

We start by performing experiments on four tasks from the DeepMind control suite: Cheetah, Hopper, Quadruped. To obtain data for the offline learning experiments we first generate a fixed dataset via a standard learning run using MPO, storing all transitions generated; we repeat this with 5 seeds for each environment. We then separate this collected data into two sets: for experiments in the high data regime, we use the first 10,000 episodes generated from each seed. For experiments with low-quality data we use the first 2,000 episodes from each seed. The high data regime therefore has both more data and data from policies which are of higher quality on average. A plot showing the performance of the initial training seeds over episodes is given in the appendix, Figure 6.

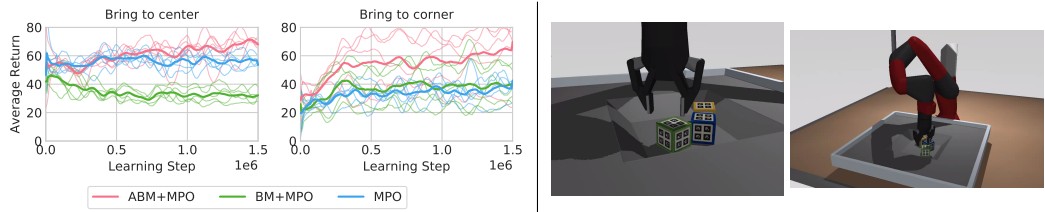

Figure 3: Learning curves for MPO with and without behavior extraction priors. We show 4 of the 7 tasks here. Five training runs are shown for each parameter setting, evaluation is average reward over 50 episodes. Behavioral modelling priors significantly improve performance across all tasks, and ABM further improves learning speed and final performance.

Figure 4: (Left) Learning curves for the tasks "bring to corner" and "bring to center". These tasks were learned using only data from the seven intial stacking tasks. The stacking dataset was rich enough to learn these new tasks fully offline with ABM. (Right) Simulated Sawyer environment.

For our offline learning experiments, we reload this data into a replay buffer.[1] The dataset is then fixed and no new transitions are added; the offline learner never receives any data that any of its current or previous policies have generated. We evaluate performance by concurrently testing the policy in the environment. The results of this evaluation are shown in Figure 1. As can be observed, standard off-policy RL algorithms (MPO / SVG) can learn some tasks offline with enough data, but learning is unstable even on these relatively simple control suite tasks – confirming previous findings from (Fujimoto et al., 2018; Kumar et al., 2019). In contrast the other methods learn stably in the high-data regime, with BCQ lagging behind in Hopper and Quadruped (sticking too close to the VAE prior actions, an effect already observed in (Kumar et al., 2019)). Remarkably our simple method of combining a policy iteration loop with a behavior model prior (BM+MPO in the plot) performs as well or better than the more complex baselines (BEAR and BCQ from the literature). Further improvement can be obtained using our advantage weighted behavior model even in some of these simple domains (ABM+SVG and ABM+MPO). Comparing the performance of the priors (BM[prior] vs ABM[prior], dotted lines) on Hopper we can understand the advantage that ABM has over BM: the BM prior performs well on simple tasks, but struggles when the data contains conflicting trajectories (as in Hopper, where some of the seeds learn sub-optimal jumping) leading to too hard constraints on the RL policy. Interestingly, the ABM prior itself performs as well or better than the baseline methods for the control-suite domains. Furthermore, in additional experiments presented in the appendix we find that competitive performance can be achieved, in simple domains, when training only an ABM prior (effectively setting $\epsilon = 0$); providing an even simpler method when one does not care about squeezing out every last bit of performance. A test on lower quality data, Figure 2, shows similar trends, with our method learning to perform slightly better than the best trajectories in the data.

## 5.2 SIMULATED ROBOT EXPERIMENTS

We experiment with a Sawyer robot arm simulated in Mujoco (Todorov et al., 2012) in a multi-task setting – as described above. The seven tasks are to manipulate blocks that are placed in the workspace of the robot. They include: reaching for the green block (Reach), grasping any block (Grasp), lifting the green block (Lift), hovering the green block over the yellow block (Place Wide), hovering the green block over the center of the yellow block (Place Narrow), stacking the green block on top of

---

[1]Due to memory constraints, it can be prohibitive to store the full dataset in replay. To circumvent this problem we run a set of "restorers", which read data from disk and add it back into the replay in a loop.

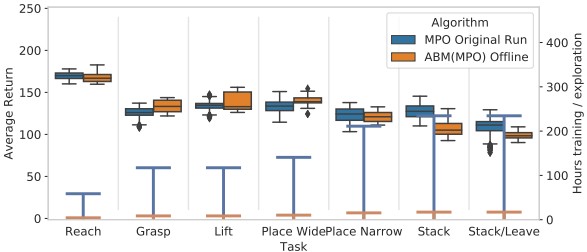 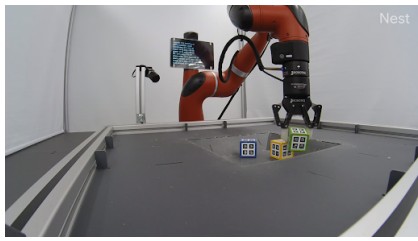

Figure 5: (Left) Original and offline learning on the real robot. The original training run takes over 200 hours to learn to stack and leave (blue bar, right y-axis) – bottle-necked by slow real-robot episode generation. Using the complete dataset of interactions in a batch-RL setting with the ABM prior, we are able to re-learn the final task in 12 hours purely from logged data (orange bar, right y-axis), while obtaining a policy that achieves similar performance on all tasks (box plots, distribution of returns over 50 test episodes, orange vs blue, left y-axis). (Right) Real Sawyer environment.

yellow (Stack and Stack/Leave i.e. without gripper contact). To generate the data for this experiment we again run MPO – here simultaneously learning all task-conditional policies for the full seven tasks. Data-was collected by randomly switching tasks after each episode (of 200 control steps) with random resets of the robot position every 20 episodes. As before, data from all executed tasks is collected in one big data-set annotating each trajectory snippet with all rewards (i.e. this is similar to the SAC-R setting from (Riedmiller et al., 2018).

During offline learning we then compare the performance of MPO and RL with a behavior modelling prior (BM+MPO and ABM+MPO). As shown in Figure 3, behavioral modelling priors improve performance across all tasks over standard MPO – which struggles in these more challenging tasks. This is likely due to the sequential nature of the tasks: later tasks implicitly include earlier tasks but only a smaller fraction of trajectories achieve success on stack and leave (and the actions needed for stack conflict, e.g., with lifting), this causes the BM prior to be overly broad (see plots in appendix). The ABM+MPO, on the other hand, achieves high performance across all tasks. Interestingly, even with ABM in place, the RL policy learned using this prior still outperforms the prior, demonstrating that RL is still useful in this setting.

As an additional experiment we test whether we can learn new tasks entirely from previously recorded data. Since our rewards are specified as functions of observations, we compute rewards for two new tasks (bringing the green block to the center and bringing it to the corner) for the entire dataset – we then test the resulting policy in the simulator. As depicted in Figure 4, this is successful with ABM+MPO, demonstrating that we can learn tasks which were not originally executed in the dataset (as long as trajectory snippets that lead to successful task execution are contained in the data).

## 5.3    REAL ROBOT EXPERIMENTS

Finally, to validate that our approach is a feasible solution to performing fast learning for real-robot experiments, we perform an experiment using a real Sawyer arm and the same set of seven tasks (implemented on the real robot) from Section 5.2. As before, data from all executed tasks is collected in one big data-set annotating each trajectory snippet with all rewards. The full buffer after about two weeks of real robot training is used as the data for offline learning; which we here only performed with ABM+MPO due to the costly evaluation. The goal is to re-learn all seven original tasks. Figure 5 shows the results of this experiment – we ran an evaluation script on the robot, continuously testing the offline learned policy, and stopped when there was no improvement in average reward (as measured over a window of 50 episodes). As can be seen, ABM with MPO as the optimizer manages to reliably re-learn all seven tasks purely from the logged data in less than 12 hours. All tasks can jointly be learned with only small differences in convergence time – while during the initial training run the harder tasks, of course, took the most time to learn. This suggests that gathering large data-sets of experience from previous robot learning experiments, and then quickly extracting the skills of interest, might be a viable strategy for making progress in robotics.

## 6 CONCLUSION

In this work, we considered the problem of stable learning from logged experience with off-policy RL algorithms. Our approach consists of using a learned prior that models the behavior distribution contained in the data (the advantage weighted behavior model) towards which the policy of an RL algorithm is regularized. This allows us to avoid drawing conclusions for which there is no evidence in the data. Our approach is robust to large amounts of sub-optimal data, and compares favourably to strong baselines on standard continuous control benchmarks. We further demonstrate that our approach can work in challenging robot manipulation domains – learning some tasks without ever seeing a single trajectory for them.

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

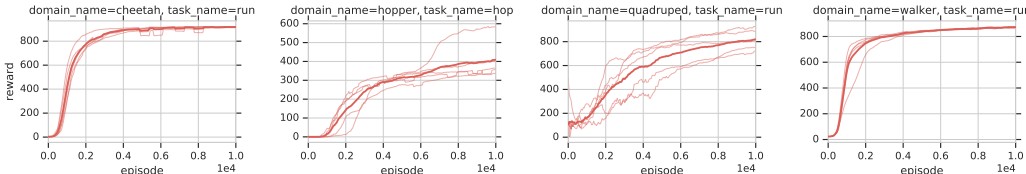

Figure 6: Learning curves for five seeds of training with MPO on the control suite tasks. Data generated during these training runs is later used for offline learning.

## A   ALGORITHM

A full algorithm listing for our procedure is given in Algorithm 1.

---

**Algorithm 1** RL with Behavior Extraction Priors

---

**Input:** $N_{steps}$ number of learning steps, $N_{\text{TU}}$ steps between target update, $M$ number of action samples, KL regularization parameter $\epsilon$, initial parameters for $\theta$, $\eta$, $\alpha$ and $\phi$
initialize $N = 0$, $\theta' = \theta$, $\phi' = \phi$
**while** $i \leq N_{\text{steps}}$ **do**
    sample a batch $\mathcal{B}$ of trajectories $\tau$ from replay buffer $\mathcal{D}_\mu$
    sample $M$ actions from policies to estimate expectations below
    // compute gradient for prior model
    **For BM prior:**
        $\delta_{\theta_{\text{prior}}} \leftarrow \nabla_{\theta_{\text{prior}}} \sum_{\tau \in \mathcal{B}} \sum_{(s_t, a_t) \in \tau} \log \pi_{\theta_{\text{bm}}}(a_t | s_t)$
    **For ABM prior:**
        $\delta_{\theta_{\text{prior}}} \leftarrow \nabla_{\theta_{\text{abm}}} \sum_{\tau \in \mathcal{B}} \sum_{(s_t, a_t) \in \tau} 1_+ (R(\tau_{1:|\tau|}) - \hat{V}_{\phi'}(s_t)) \log \pi_{\theta_{\text{abm}}}(a_t | s_t)$
    // compute gradients for $Q$, $\pi$ and $\eta$
    $\delta_\phi \leftarrow \nabla_\phi \sum_{\tau \in \mathcal{B}} \sum_{i \sim I} \sum_{(s_t, a_t) \in \tau} \left( r(s_t, a_t) + \gamma \hat{V}_{\phi'}(s_t) - \hat{Q}_\phi(s_t, a_t) \right)^2$
    **For MPO:**
        $\delta_\pi \leftarrow -\nabla_\theta \sum_{\tau \in \mathcal{B}} \sum_{s_t \in \tau} \mathbb{E}_{a \sim \pi_{\text{prior}}} \left[ \exp\left(\hat{Q}_{\phi'}(s_t, a)/\eta\right) \log \pi_\theta(a|s_t) \right] + \alpha(\epsilon_{\text{trust}} - KL[\pi_{\theta'} \| \pi_\theta])$
        $\delta_\alpha \leftarrow \nabla_\alpha \sum_{\tau \in \mathcal{B}} \sum_{s_t \in \tau} \alpha(\epsilon_{\text{trust}} - KL[\pi_{\theta'} \| \pi_\theta])$
        $\delta_\eta \leftarrow \nabla_\eta g(\eta) = \sum_{\tau \in \mathcal{B}} \nabla_\eta \eta \epsilon + \eta \sum_{s_t \in \tau} \log \mathbb{E}_{a \sim \pi_{\text{prior}}} \left[ \exp\left(\hat{Q}_{\phi'}(s_t, a_j)/\eta\right) \right]$
    **For Stochastic Value Gradient:**
        $\delta_\pi \leftarrow -\sum_{\tau \in \mathcal{B}} \nabla_\theta \sum_{s_t \in \tau} \mathbb{E}_{a \sim \pi_\theta(\cdot|s)} \left[ \hat{Q}_{\phi'}(s_t, a) \right] + (\epsilon - \eta KL[\pi_\theta \| \pi_{\text{prior}}])$
        $\delta_\eta \leftarrow \sum_{\tau \in \mathcal{B}} \nabla_\eta \sum_{s_t \in \tau} \eta(\epsilon - KL[\pi_\theta \| \pi_{\text{prior}}])$
        $\delta_\alpha \leftarrow 0$ // not used
    // apply gradient updates
    $\theta, \theta_{\text{prior}}, \eta, \alpha, \phi = \text{optimizer\_update}([\theta, \theta_{\text{prior}}, \eta, \alpha, \phi], [\delta_\pi, \delta_{\theta_{\text{prior}}}, \delta_\eta, \delta_\alpha, \delta_\phi])$,
    $i = i + 1$
    // update target networks
    **if** $i \mod N_{\text{TU}} = 0$ **then**
        $\theta' = \theta$, $\phi' = \phi$
    **end if**
**end while**

---

## B   DETAILS ON POLICY IMPROVEMENT

We here give additional details on the implementation of the policy improvement step in our algorithm. Depending on the policy optimizer used (MPO or SVG) different update rules are used to maximize the objective given in Equation 4. This is also outlined in Algorithm 1. We here describe the general form for both algorithms, using $\pi_{\text{prior}}$ to represent the prior which then will be instantiated as either the ABM or BM prior.

## B.1 MPO

For the EM-style optimization based on MPO we first notice that the optimal non-parametric policy that respects the KL constraint wrt. $\pi_{\text{prior}}$ is

$$\hat{\pi}(a|s) = \pi_{\text{prior}}(a|s) \exp(\hat{Q}^{\pi_i}(s,a_j)/\eta - \log Z), \tag{7}$$

with $Z = \int \pi_{\text{prior}}(a|s) \exp(\hat{Q}^{\pi_i}(s,a)/\eta) da$ and where $\eta$ is a temperature that depends on the desired constraint $\epsilon$. In practice we estimate $Z$ based on the $M$ samples $\{a_1, \dots, a_M\} \sim \pi_{\text{prior}}(a|s)$ that we draw for each state to perform the optimization of $\pi_\theta$. That is we set $Z \approx 1/M \sum_{j=1}^M \exp(\hat{Q}^{\pi_i}(s,a_j)/\eta)$. Using these samples we can then optimize for $\eta$ in a way analogous to what is described in Abdolmaleki et al. (2018). Specifically, we find that the objective for finding $\eta$ is

$$g(\eta) = \mathbb{E}_{\tau \in \mathcal{D}_\mu} \Big[ \eta\epsilon + \eta \sum_{s_t \in \tau} \log \mathbb{E}_{a \sim \pi_{\text{prior}}} \big[ \exp \big( \hat{Q}_{\phi'}(s,a)/\eta \big) \big] \Big], \tag{8}$$

which can approximate based on a batch $\mathcal{B}$ of trajectories sampled from $\mathcal{D}_\mu$ (sampling $M$ actions from $\pi_{\text{prior}}$ for each state therein) and corresponding action samples where we used the samples $\{a_1, \dots, a_M\} \sim \pi_{\text{prior}}(a|s)$ as

$$g(\eta) \approx \sum_{\tau \in \mathcal{B}} \Big[ \eta\epsilon + \eta \sum_{s_t \in \tau} \log \frac{1}{M} \sum_{j=1}^M \exp \big( \hat{Q}_{\phi'}(s,a_j)/\eta \big) \Big], \tag{9}$$

which can readily be differentiated wrt. $\eta$. We then use Adam (with standard settings and learning rate $2e-4$) to take a gradient step in direction of $\nabla_\eta g(\eta)$ (we want to maximize $g(\eta)$) for each batch. We start our optimization with $\eta = 3$ to ensure stable optimization (i.e. to avoid large changes in the policy parameters in the beginning of optimization). Further, after each gradient step, we project $\eta$ to the positive numbers i.e. we set $\eta = \max(\eta, 0.001)$, as $\eta$ is required to be positive. We find that this procedure is capable of fulfilling the desired KL constraints well.

The same batch, and action samples, are then also used to take an optimization step for the policy parameters $\theta$. In particular we find the parametric policy by minimizing the divergence $KL(\hat{\pi} \| \pi_{\theta_{i+1}})$, which is equivalent to maximizing the weighted log likelihood of sampled actions (Equation 5). We only take $M = 20$ samples here, which is a relatively crude representation of the behavior model at state $s$; therefore, to prevent the policy from converging too quickly it can be useful to employ an additional trust region constraint in this step that ensures slow convergence. We do this by adjusting the maximum likelihood objective Equation 5 to contain an additional KL regularization towards the previous policy, yielding the following optimization problem using samples $\{a_1, \dots, a_M\} \sim \pi_{\text{prior}}(a|s)$:

$$\max_\theta \min_\alpha \sum_{\tau \in \mathcal{B}} \sum_{s_t \in \tau} \sum_{j=1}^M \Big[ \exp(\hat{Q}^{\pi_i}(s_t,a_j)/\eta) \log \pi_\theta(a_j|s_t) + \alpha(\epsilon_{\text{trust}} - KL(\pi_{\theta_i} \| \pi_\theta)) \Big], \tag{10}$$

for $\alpha > 0$, which is a Langrangian relaxation to the maximum likelihood problem under the additional constraint that $KL(\pi_{\theta_i} \| \pi_\theta) < \epsilon_{\text{trust}}$, where $\alpha$ is the Langrange multiplier. This objective can be differentiated wrt. both $\theta$ and $\alpha$ and we simply take alternating gradient descent steps (one per batch for both $\theta$ and $\alpha$) using Adam (Kingma & Ba, 2015) (starting with a random $\theta_0$ and $\alpha = 1$) and projecting $\alpha$ back to the positive regime if it becomes negative; i.e. we set $\alpha = max(\alpha, 0.001)$.

## B.2 SVG

To optimize the policy parameters $\theta$ via the stochastic value gradient (Heess et al., 2015) (under a KL constraints) we can directly calculate the derivative of the Langrangian relaxation from Equation 6. In particular, again assuming that we have sampled a batch $\mathcal{B}$, the gradient wrt. $\theta$ can be obtained via the reparameterization trick (Kingma & Welling, 2014; Rezende et al., 2014). For this we first require that a sample from our policy $\pi_\theta(a|s)$ can be obtained via a deterministic function applied to a standard noise source. We first specify the policy class used in the paper to be that of Gaussian policies, parameterized as $\pi_\theta(a|s) = \mathcal{N}(a|\mu_\theta(s), I\sigma_\theta^2(s))$, where $\mathcal{N}(a|\mu, \Sigma)$ denotes the pdf of a standard Normal distribution and where we assume $\mu_\theta$ is directly given as one output of the network whereas we parameterize the diagonal standard deviation as $\sigma_\theta(s) = softplus(h_\theta(s))$ with $h_\theta(s)$

being output by the neural network. We can then obtain samples via the deterministic transformation $f(s, \xi; \theta) = \mu_\theta(s) + \sigma_\theta(s)\xi$, where $\xi \sim \mathcal{N}(0, I)$ ($I$ being the identity matrix). Using this definition we can obtain the following expression for the value gradient:

$$
\begin{aligned}
\delta_\theta =& \nabla_\theta \mathop{\mathbb{E}}_{\tau \sim \mathcal{D}_\mu} \Big[ \mathbb{E}_{a \sim \pi_\theta(\cdot|s)} \big[ \hat{Q}^{\pi_i}(s, a) \big] + \eta(\epsilon - \text{KL}[\pi_\theta(\cdot|s) \| \pi_{\text{prior}}(\cdot|s)]) \Big] \\
=& \sum_{\tau \in \mathcal{B}} \sum_{s_t \in \tau} \nabla_\theta \Big[ \mathbb{E}_{a \sim \pi_\theta(\cdot|s)} \big[ \hat{Q}^{\pi_i}(s, a) \big] + \eta(\epsilon - \text{KL}[\pi_\theta(\cdot|s) \| \pi_{\text{prior}}(\cdot|s)]) \Big] \\
=& \sum_{\tau \in \mathcal{B}} \sum_{s_t \in \tau} \frac{1}{M} \sum_{j=1}^{M} \Big[ \nabla_\theta f(s, \xi_j; \theta) \nabla_f Q(s, f(s, \xi_j; \theta)) \Big] + \nabla_\theta \eta(\epsilon - \text{KL}[\pi_\theta(\cdot|s) \| \pi_{\text{prior}}(\cdot|s)]),
\end{aligned}
$$
(11)

where we use the Gaussian samples $\{\xi_1, \dots, \xi_M\} \sim \mathcal{N}(0, I)$. The gradient for the Langrangian multiplier is given as

$$
\begin{aligned}
\delta_\eta =& \nabla_\eta \mathop{\mathbb{E}}_{\tau \sim \mathcal{D}_\mu} \Big[ \mathbb{E}_{a \sim \pi_\theta(\cdot|s)} \big[ \hat{Q}^{\pi_i}(s, a) \big] + \eta(\epsilon - \text{KL}[\pi_\theta(\cdot|s) \| \pi_{\text{prior}}(\cdot|s)]) \Big] \\
=& \sum_{\tau \in \mathcal{B}} \sum_{s_t \in \tau} \nabla_\eta \eta(\epsilon - \text{KL}[\pi_\theta(\cdot|s) \| \pi_{\text{prior}}(\cdot|s)]),
\end{aligned}
$$
(12)

where we dropped terms independent of $\eta$ in the second line. Following $\delta_\theta$ to maximize the objective, and conversely moving in the opposite direction of $\delta_\eta$ to minimize the objective wrt. $\eta$, can then be performed by taking alternating gradient steps. We perform one step per batch for both $\eta$ and $\theta$ via Adam; starting from an initially random $\theta$ and $\eta = 1$. As in the MPO procedure we ensure that $\eta$ is positive by projecting it to the positive regime after each gradient step.

## C  DETAILS ON THE EXPERIMENTAL SETUP

### C.1  HYPERPARAMETERS AND NETWORK ARCHITECTURE

Hyperparameters for our MPO, SVG, and BCQ single-task experiments are shown in tables 1, 2, and 3, respectively. For multitask experiments, we modify the parameters shown in 5.

### C.2  DETAILS ON BCQ AND BEAR

To provide strong off-policy learning baselines we re-implemented BCQ (Fujimoto et al., 2018) and BEAR (Kumar et al., 2019) in the same framework that we used to implement our own algorithm. As mentioned in the main paper we used the same network architecture for ll algorithms. Algorithm specific hyperparameters where tuned via a coarse grid search on the control suite tasks; while following the advice from the original papers on good parameter ranges. To avoid bias in our comparisons we did not utilize ensembles of Q-functions for any of the methods (e.g. we removed them from BEAR), we note that ensembling did not seem to have a major impact on performance (see appendix in (Kumar et al., 2019)). Parameters for all methods where optimized with Adam. Furthermore, to apply BEAR and BCQ in the multi-task setting we employed the same conditioning of the policy and Q-function on a one-hot task vector (which is used to select among multiple network "heads", yielding per task parameters, see description below).

For BCQ we used a range of $[0.25, 0.25]$ for the perturbative actions generated by the DDPG trained network, and chose a latent dimensionality of 64 for the VAE, see Table 3 for the full hyperparameters.

For BEAR we used a KL constraint rather than the maximum mean discrepancy. This ensures comparability with our method and we did not see any issues with instability when using a KL. To ensure good satisfaction of constraints we used the exact same optimization for the Langragian multiplier required in BEAR that was also used for our method – see description of SVG above. The hyperparameters for the BEAR training run on the control suite are given in Table 4.

### C.3  DETAILS ON THE ROBOT EXPERIMENT SETUP

The task setup for both the simulated and real robot experiments is described in the following. A detailed description of the robot setup will be given in an accompanying paper. We nonetheless give

| Hyperparameters | MPO |
|---|---|
| Policy net | 256-256 |
| Prior net | 256-256 |
| Number of actions sampled per state | 20 |
| Q function net | 256-256-256 |
| $\epsilon$ | 0.1 |
| $\epsilon_\mu$ | $5 \times 10^{-3}$ |
| $\epsilon_\Sigma$ | $1 \times 10^{-5}$ |
| Discount factor ($\gamma$) | 0.99 |
| Adam learning rate | $2 \times 10^{-4}$ |
| Replay buffer size | $2 \times 10^{6}$ |
| Target network update period | 200 |
| Batch size | 512 |
| Activation function | elu |
| Layer norm on first layer | Yes |
| Tanh on Gaussian mean | No |
| Min variance | 0.01 |
| Max variance | unbounded |

Table 1: Hyperparameters for MPO

| Hyperparameters | SVG |
|---|---|
| Policy net | 256-256 |
| Prior net | 256-256 |
| Q function net | 256-256-256 |
| $\epsilon$ | 0.2 |
| Discount factor ($\gamma$) | 0.99 |
| Adam learning rate | $2 \times 10^{-4}$ |
| Replay buffer size | $2 \times 10^{6}$ |
| Target network update period | 200 |
| Batch size | 512 |
| Activation function | elu |
| Layer norm on first layer | Yes |
| Tanh on Gaussian mean | No |
| Min variance | 0.01 |
| Max variance | unbounded |

Table 2: Hyperparameters for SVG

a description here for completeness. We make no claim to have contributed these tasks specifically for this paper and merely use them as an evaluation test-bed.

As the robot we utilize a Sawyer robotic arm mounted on a table and equipped with a Robotiq 2F-85 parallel gripper. A basket is positioned in front of the robot which contains three cubes (the proportions of cubes and basket sizes are consistent between simulation and reality). Three cameras on the basket track the cube using augmented reality tags. To model the tasks as an MDP we provide both proprioceptive information from the robot sensors (joint positions, velocities and torques) and the tracked cube position, velocity (both in 3 dimensions) and orientation to the policy and Q-function. Overall the observations provided to the robot are: Proprioception: Joint positions (7D, double), velocities (7D, double), torques (7D, double); wrist pose (7D, double), velocity (6D, double), force (6D, double); gripper finger angles (1D, int), velocity (1D, int), grasp flag (1D, binary) and the object features: Object pose (7D, double) averaged over all cameras observing the object, Relative pose between object and gripper (7D, double). In simulation the true object position and velocities / orientation are used instead of running a tracking algorithm.

We control both the robot and the gripper by commanding velocities in the 4-dimensional Cartesian space of the robot's endeffector (specifying three translational velocities plus the velocity for the wrists rotation) while the gripper control is 1-dimensional. The action limits are [-0.07, 0.07] m/s

| Hyperparameters | BCQ |
|---|---|
| Encoder net | 256-256 |
| Latent size | 64 |
| Decoder net | 256-256 |
| Perturbation net | 256-256 |
| Q function net | 256-256-256 |
| Perturbation Scale Factor | 0.25 |
| Discount factor ($\gamma$) | 0.99 |
| Adam learning rate | $2 \times 10^{-4}$ |
| Replay buffer size | $2 \times 10^{6}$ |
| Batch size | 512 |

Table 3: Hyperparameters for BCQ

| Hyperparameters | BEAR |
|---|---|
| Policy net | 256-256 |
| Prior net | 256-256 |
| Q function net | 256-256-256 |
| $\epsilon$ for KL constraint | 0.2 |
| action samples for BEAR-QL | 20 |
| Discount factor ($\gamma$) | 0.99 |
| Adam learning rate | $2 \times 10^{-4}$ |
| Replay buffer size | $2 \times 10^{6}$ |
| Target network update period | 200 |
| Batch size | 512 |
| Activation function | elu |
| Layer norm on first layer | Yes |
| Tanh on Gaussian mean | No |
| Min variance | 0.01 |
| Max variance | unbounded |

Table 4: Hyperparameters for BEAR

for Cartesian actions, [-1, 1] rad/s for wrist rotation and [-255, 255] for finger velocity (for units see gripper specifications). The control rate at which the actions are executed is 20 Hz. Episodes are terminated if the wrist force exceeds 20 N on any axis, encouraging a gentle interaction of the robot with its environment.

For our experiment we use 7 different task to learn. The first 6 tasks are auxiliary tasks for better exploration that help to learn the final task (Stack and Leave), i.e, stacking the green cube on top of the yellow cube. The rewards functions for all tasks are given as:

1. *Reach*: $stol(d(\text{pos}_{\text{endeffector}}, \text{pos}_{\text{green}}), 0.02, 0.15)$:
   This function minimizes the distance $d$ of the gripper to the green cube.

2. *Grasp*:
   Activate grasp sensor of gripper.

3. *Lift*: 0 if height of green block is less than 3 cm, if it is above 10 centimeters and $(height - 3cm)/(10cm - 3cm)$ otherwise.

4. *Place Wide*: $stol(d(\text{pos}_{\text{green}}, \text{pos}_{\text{yellow}} + [0, 0, 0.05]), 0.01, 0.20)$
   Move green cube to a position 5cm above the yellow cube.

5. *Place Narrow*: $stol(d(\text{pos}_{\text{green}}, \text{pos}_{\text{yellow}} + [0, 0, 0.05]), 0.00, 0.01)$:
   Like Place Wide but with more precision.

6. *Stack*: $btol(d_{xy}(\text{pos}_{\text{green}}, \text{pos}_{\text{yellow}}), 0.03) * btol(d_z(\text{pos}_{\text{green}}, \text{pos}_{\text{yellow}}) + 0.05, 0.01) * (1 - Grasp)$
   A binary reward for stacking green cube on the yellow one and deactivating the grasp sensor.

| Hyperparameters | Multitask |
|---|---|
| Policy net | 200 -> 300 |
| Q function net | 400 -> 400 |
| Encoder net (BCQ) | 256 -> 256-256 |
| Decoder net (BCQ) | 256-256 |
| Perturbation net (BCQ) | 256 -> 100 |
| Replay buffer size | $4 \times 10^6$ |

Table 5: Network parameters for multitask experiments. "->" indicates the network branching into separate "heads" for each task, which are selected based on the one-hot task vector. This architecture is similar to the design presented in (Riedmiller et al., 2018). Note that transitions are duplicated in replay for each task (so that transitions are sampled independently for each task).

7. *STACK_AND_LEAVE(G, Y)*: $stol(d_z(\text{pos}_{\text{endeffector}}, \text{pos}_{\text{green}}) + 0.10, 0.03, 0.10) * Stack$
   Like STACK(G, Y), but it gets max reward if it moves the arm 10cm above the green cube.

Where $d(a, b)$ denotes the euclidean distance between a and b. We also define two tolerance functions with outputs scaled between 0 and 1, i.e,

$$stol(v, \epsilon, r) = \begin{cases} 1 & \text{iff } |v| < \epsilon \\ 1 - tanh^2(\frac{atanh(\sqrt{0.95})}{r}|v|) & \text{else,} \end{cases} \tag{13}$$

$$btol(v, \epsilon) = \begin{cases} 1 & \text{iff } |v| < \epsilon \\ 0 & \text{else.} \end{cases} \tag{14}$$

# D  ADDITIONAL EXPERIMENTAL RESULTS

We present additional plots that show some aspects of the developed algorithm in more detail.

## D.1  EXPANDED PLOTS FOR CONTROL SUITE

We provide expanded plots for MPO on the control suite in Figure 7. These show in detail that the learned advantage weighted behavior model (ABM, in red in the left column) is far superior to the standard behavior model prior, leading to less constrained RL policies.

## D.2  EXPANDED PLOTS FOR ROBOT SIMULATION

Figure 8 shows full results for the simulated robot stacking task, including all 7 intentions as well as performance of the prior policies themselves during learning. The task set is structured: earlier tasks like reaching and lifting are necessary to perform most other tasks, so the simple behavioral model performs well on these. For the more difficult stacking tasks, however, the presence of conflicting data means the simple behavioral model doesn't achieve high reward, though it still significantly improves performance of the regularized policy.

## D.3  PERFORMANCE TABLES FOR CONTROL SUITE AND ROBOT SIMULATION

Table 6 shows final performance for all methods on control suite tasks. Table 7 shows final performance on simulated robotics tasks to make comparison between algorithms easier. The episode returns are averaged over the final 10% of episodes. ABM provides a performance boost over BM alone, particularly for difficult tasks such as block stacking and quadruped. The RL policy further improves performance on the difficult tasks. As noted in the main paper, one additional option, to further simplify the algorithm is to omit the policy improvement step, setting $\pi_{i+1} = \pi_{\text{prior}}$ (i.e. considering the case where $\epsilon = 0$) and, conversely learning the Q-values of the prior. In additional experiments we have found that this procedure roughly recovers the performance of the ABM prior when trained together with MPO (ABM+MPO); i.e. this is an option for a simpler algorithm to

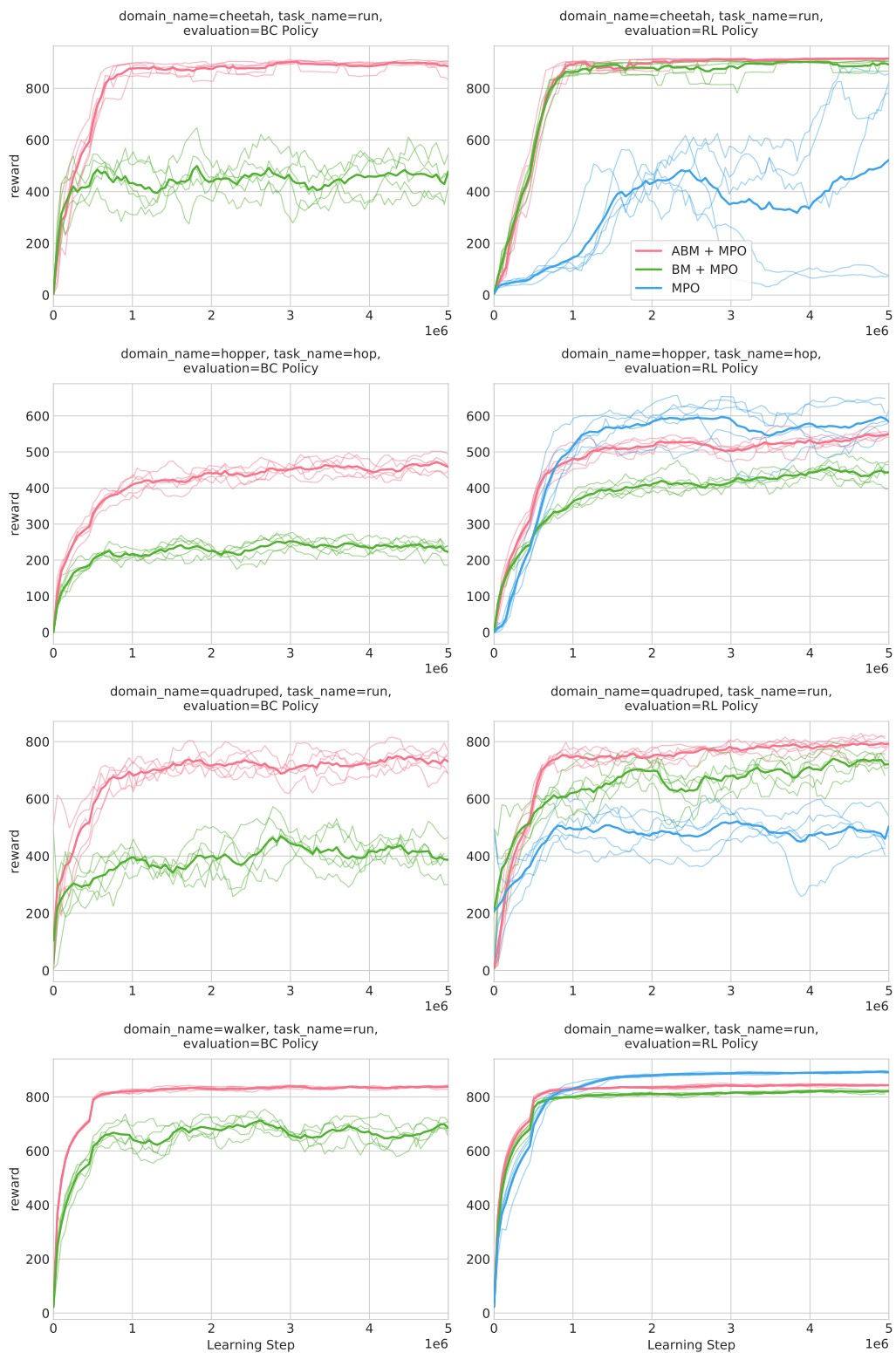

Figure 7: Detailed plots for the control suite showing the performance of the respective learned prior (left column) and RL policy (right column) for all tasks (one per row).

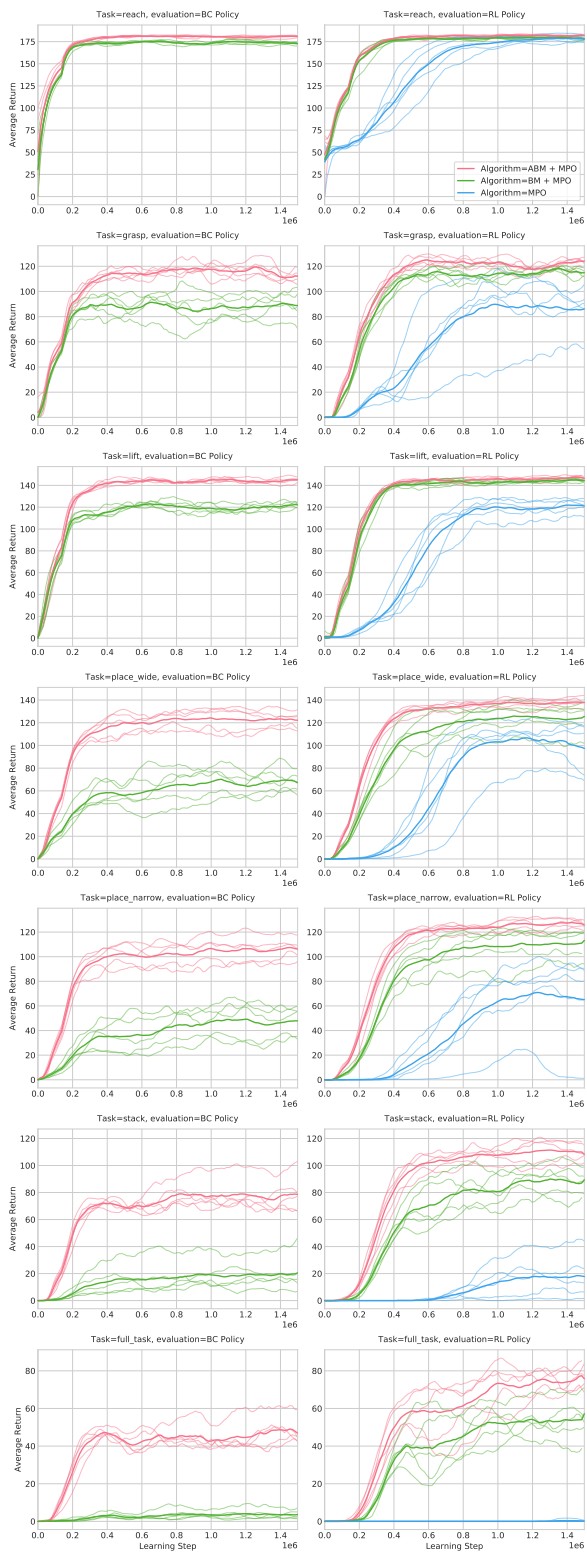

Figure 8: All intentions for blocks stacking in simulation. Performance of executing the prior on the left, performance of the algorithm learned using this prior on the right.

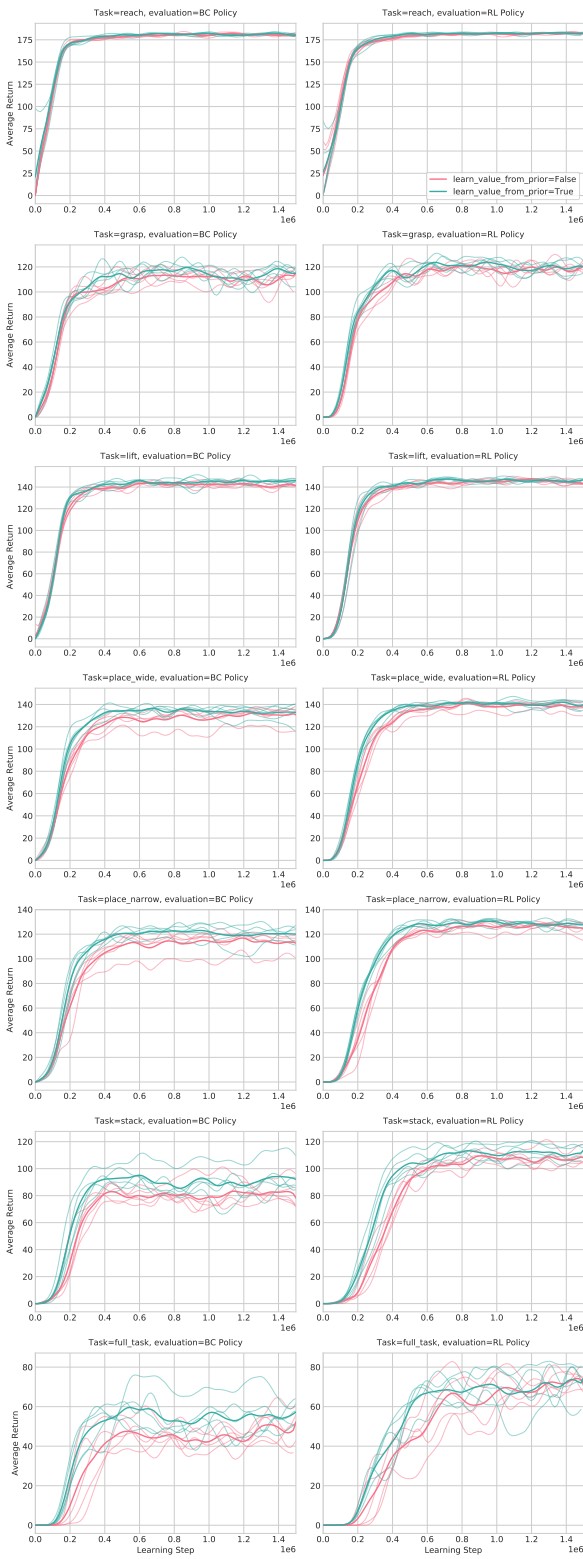

Figure 9: Learning the value function from the ABM prior policy on block stacking in simulation. When the value function is learned from the prior, the ABM prior no longer depends on the RL policy, and can be learned on its own. We can still perform MPO policy improvement using this value function and regularized to this new prior, shown on the right. Using a value function learned from the prior achieves similar performance, and may in fact perform slightly better on more difficult tasks.

| Intention \Algorithm | ABM + MPO | prior | BM + MPO | prior | MPO | ABM + SVG | BM + SVG | SVG | BCQ | BEAR | ABM ($\epsilon = 0, |\tau| = 2$) |
|---|---|---|---|---|---|---|---|---|---|---|---|
| cheetah | 907.4 | 888.7 | 891.8 | 465.3 | 496.2 | 920.7 | 924.1 | 726.1 | 914.9 | 843 | 884.3 |
| hopper | 539.3 | 454.8 | 453.8 | 237.2 | 599.2 | 651.6 | 634.7 | 544.1 | 341.2 | 425 | 451.8 |
| quadruped | 786.6 | 733.4 | 710.8 | 398.8 | 465.6 | 856.5 | 818.7 | 472.7 | 621.4 | 674 | 733.6 |
| walker | 843.0 | 835.9 | 817.7 | 684.8 | 891.9 | 896.5 | 895.6 | 891.0 | 869.5 | 814 | 838.1 |

Table 6: Episode returns on control suite tasks when learning from 10k episodes – for comparison between algorithms.

| Intention \Algorithm | ABM + MPO | prior | BM + MPO | prior | MPO | BCQ | BEAR |
|---|---|---|---|---|---|---|---|
| Reach | 182.3 | 180.8 | 178.3 | 172.9 | 178.3 | 172.6 | 176.9 |
| Grasp | 123.7 | 111.7 | 114.6 | 89.1 | 86.0 | 104.7 | 116.3 |
| Lift | 147.1 | 144.9 | 144.2 | 122.5 | 121.4 | 126.4 | 137.9 |
| Place Wide | 137.9 | 122.3 | 124.4 | 67.2 | 98.0 | 68.0 | 93.2 |
| Place Narrow | 126.3 | 106.1 | 111.3 | 46.7 | 65.3 | 58.3 | 64.1 |
| Stack | 109.9 | 78.8 | 86.9 | 20.1 | 17.5 | 39.3 | 67.8 |
| Full Task | 77.9 | 47.3 | 53.5 | 3.5 | 0.2 | 5.2 | 34.6 |

Table 7: Returns on each simulated robotics task – for comparison between algorithms.

implement at the cost of some performance loss (especially on the most complicated domains). We included this setting as ABM ($\epsilon = 0, |\tau| = 2$) in the table – noting that in this case it is vital to choose short trajectory snippets in order to allow $\pi_{\text{prior}}$ to pick the best action in each state.

