# OpenReview forum: "Keep Doing What Worked: Behavior Modelling Priors for Offline Reinforcement Learning"
_ICLR.cc/2020/Conference — Accept (Poster)_

### Official Review · AnonReviewer1 · 2019-10-21
**Official Blind Review #1**

**Rating:** 6

**Review:**

This paper proposes a method for “offline RL” (a.k.a “batch RL”), i.e. reinforcement learning from a given static dataset, with no option to perform on-policy data collection. Contrary to prior work (Fujimoto et al, Kumar et al, Agarwal et al) in this area which focuses on making Q-learning robust in the offline RL setting, the authors in this paper instead propose making the policy update step robust to the batch RL setting. They achieve this by constraining the policy that is being learned to stay close to a learned “prior policy” (using KL divergence as a distance metric). The authors provide two different ways of learning this prior policy  - the first one is a simple behavior model policy (which involves fitting a model using maximum likelihood on the entire dataset - effectively performing behavior cloning on the entire dataset), while the second is a more sophisticated “advantage behavior model”, which fits a model only on the “good” data (i.e. environment transitions with a positive advantage, where the advantage is estimated using the current policy). The authors extend the MPO and SVG algorithms to work with this constrained update step.

The authors provide experiments on the standard DM control suite tasks and some robotic tasks (both simulated and real world). For all experiments, the authors collect data by first running MPO in the standard RL setting (i.e. online data collection), and use the replay buffer from these successful RL runs to relearn a policy. The proposed method performs comparable to other methods in some settings (like all the experiments in the 2K episodes setting in Figure 2), and performs slightly better in some other settings (like Hopper and Quadruped in the 10K episodes setting). The gains are more significant in the non-standard robotics domains (Stack, Stack/Leave). The real world robotics results are not compared to any baseline (perhaps due to evaluation being expensive / time-consuming). However, I think the experimental results, especially Figures 7 and 8, point to a potential flaw in the proposed algorithm. In Figure 7 in the appendix, the learned behavior model (ABM) gets about the same performance as the proposed method (ABM+MPO) in pretty much all the four tasks. In one of the tasks (hopper), ABM+MPO does slightly better than ABM alone, but in this case plane MPO is better than ABM + MPO, indicating that this might just be a task on which MPO does well (for some unexplained reason). Similarly, in the robotics results in Figure 8, ABM alone does comparable to ABM+MPO in five of the seven tasks.

These experimental results imply to me that most of the performance actually comes from the learned prior (ABM), and whether performing MPO on top of it leads to any improvement or not is hard to predict^[1]. The ABM model itself is somewhat novel, and similar ideas under review at ICLR this year have shown that this alone is a useful algorithm for offline RL, see https://openreview.net/forum?id=H1gdF34FvS and https://openreview.net/forum?id=BJlnmgrFvS. However, the ABM model is not emphasized much in the paper. It is also unclear to my as to why the ABM/BM results were placed in the Appendix for Figure 3, while similar results for experiments in Figure 1 were provided using dotted lines.

For the paper to be accepted for publication, I think it needs to make a stronger argument (experimentally, at least) about the proposed algorithm being superior to ABM. If this is not really the case, then I think it would require a substantial rewrite to emphasize that ABM is where most of the model performance comes from (as seen in concurrent work listed above).

[1]: A slight caveat here: in the proposed Algorithm 1, the ABM model is learned online along with the policy, but I believe it could be learned without the policy - for example, by using MC returns as in https://openreview.net/forum?id=H1gdF34FvS.
----------------------

Edit: The author response has convinced me to bump my rating to a weak accept.

**Experience Assessment:**

I have read many papers in this area.

**Review Assessment: Checking Correctness Of Derivations And Theory:**

I carefully checked the derivations and theory.

**Review Assessment: Checking Correctness Of Experiments:**

I carefully checked the experiments.

**Review Assessment: Thoroughness In Paper Reading:**

I read the paper at least twice and used my best judgement in assessing the paper.

---

> ### Author Response · Authors · 2019-11-15
> **Response to Review #1**
>
> We thank the reviewer for the thorough review and detailed responses.
> We want to first address the reviewers concerns regarding the benefits that learning an RL policy in addition to the ABM prior brings:
>
> 1. We agree that the idea of learning an ABM based on trajectories in the data that are “good” for the current task is one of the main contributions of the paper. We also agree that one of its major benefits is its simplicity: in particular, it requires no hyper-parameters to be tuned (like the scale of an exponential function of the Q-values / Returns). The reviewer thought that this contribution could have been emphasized more clearly and we have done so both in the methods and experimental section now (please see the updated paper).
>
> 2. Regarding the improvement RL+ABM obtains over ABM: The ABM policy is indeed often a good policy by itself. This is particularly true for simple control suite tasks such as cheetah and walker. For the most difficult tasks in our experimental comparisons (stacking with the robot arm in simulation), learning an RL policy using the prior still provides significant benefits to performance. For the two most difficult tasks (stacking the blocks and the full task, “stack-and-leave”) the ABM policy only achieves about 70 % (stack) and 60 % (stack and leave) of the RL policy performance (i.e. it often drops the block and has to try again).
> We appreciate that spotting this difference might have not been easy in the initial version of the paper (because the axes in Figures 7 and 8 in the appendix had different limits). We have updated the plots to have uniform axis scales between BM and RL policies. In addition, we’ve included a tabular representation of the algorithm performances in the appendix to make direct comparison easier.
>
> 3. We want to emphasize that all different variants of our algorithm significantly outperform the published state-of-the-art baselines (BCQ & BEAR) on the tasks we consider. We think that this is an important contribution to the literature on BatchRL.
>
> 4. Regarding prior performance plots in the main paper: We omitted these in the initial submission for some of the plots based on feedback on an early draft that the plots were “too busy”. We appreciate that this omission was more confusing than helpful to the reviewer and have included dashed lines - indicating the performance of the prior - in all plots in the main paper now.
>
> We also understand that it can be appealing to have an even simpler algorithm (removing the policy improvement step as suggested by the reviewer) when one does not care about top performance. As the reviewer points out, this can be expected to give slightly different results to an ABM prior learned alongside the RL policy (because the advantage calculated in that case is wrt. the RL policy). We have performed initial experiments in this direction which indicate that simply regressing monte-carlo returns in combination with the ABM prior (i.e. not learning a Q-function but regressing a value V(x) against n-step bootstrapped returns) works ok for control suite domains (as perhaps also indicated by the concurrent submissions mentioned by the reviewer), but fails for the multi-task robot data. This highlights a feature of our paper: while prior work considered learning from “noisy” data collected from either random or partially-trained policies, we consider multi-task experiments where one must learn from data generated by policies with goals that differ in non-random ways. While MC returns do not work well (due to the noise in the return estimation), another option is to stick to the policy iteration scheme outlined in our paper, but remove the policy improvement step and set pi_{i+1} = pi_\prior, i.e. assuming \epsilon = 0 - thus learning a Q-function for the ABM prior. We find that with this procedure we can train an ABM model that is about as good as the ABM prior obtained during learning of ABM+MPO. In short: this does give a further simplified method, though it doesn’t achieve the maximum performance on the more difficult tasks (as described above). We have included a discussion in the appendix and have added results for the control suite. We will add the corresponding results for the simulated robot domain once training finishes, though ongoing experiments suggest similar results (i.e. ABM recovers the prior performance).
>
> Finally, regarding the question about baselines for the real robot experiment: Indeed the initial run for collecting the data took over a week, training offline while recording all statistics on the real robot took another 3 days (the robot was used only for evaluation purposes during that time). We hence settled on comparing different algorithms in simulation, which we found in previous experiments to be a good predictor of real robot performance.

---

### Official Review · AnonReviewer3 · 2019-10-22
**Official Blind Review #3**

**Rating:** 6

**Review:**

This paper proposes a novel off-policy reinforcement learning algorithm based on a learned prior. Although it is just an extension of existing works, it does outperform the-state-of-the-art methods and can be used in real-world robots.The technique details are well introduced and analysed. The experimental results and comparison are sufficient.

However, there are also some details require further explanation. For example, In Equation (10), the Langrange multiplier should be constrained to non-negative. But the authors have not taken it into consideration in the optimization process.
Besides, there are also some typesetting problems in the paper, such as in Fig. 7.


**Experience Assessment:**

I have read many papers in this area.

**Review Assessment: Checking Correctness Of Derivations And Theory:**

I assessed the sensibility of the derivations and theory.

**Review Assessment: Checking Correctness Of Experiments:**

I carefully checked the experiments.

**Review Assessment: Thoroughness In Paper Reading:**

I read the paper at least twice and used my best judgement in assessing the paper.

---

> ### Author Response · Authors · 2019-11-15
> **Response to Review #3**
>
> We thank the reviewer for the review and comments as well as spotting two small errors.
> Indeed the reviewer is correct, the Lagrange multiplier should be non-negative. This is handled correctly in our implementation but was not explained in the appendix for this step. We have updated the paper to reflect this.
>
> We have also fixed the mentioned issue with overlapping text in Fig. 7, and we have used consistent axis scales between prior and RL plots in the appendix to make scale comparisons clearer. In addition we included tables in the appendix to make comparison easier and updated the plots in the main paper.

---

### Official Review · AnonReviewer2 · 2019-10-23
**Official Blind Review #2**

**Rating:** 6

**Review:**

This paper present a novel approach to reinforcement learn from batched data that come from very different sources. To achieve this, they propose to learn a prior model and then constrain the RL policy to a trust-region that does not deviate from the domain where the current policy is close to.

The paper is clearly written.

Strength: the paper is very novel and tries to solve a very challenging problem. The success of this approach shows the potential that collecting large-scale data without distinguishment is possible. With that, the data efficiency will be much improved! The experiment itself shows superior performance than other methods. This method also works for real robots.

Weakness: There is no algorithm box so it can be hard to fully reproduce. The paper mentions stableness for the method however the analysis on this aspect is limited.

**Experience Assessment:**

I have published one or two papers in this area.

**Review Assessment: Checking Correctness Of Derivations And Theory:**

I carefully checked the derivations and theory.

**Review Assessment: Checking Correctness Of Experiments:**

I carefully checked the experiments.

**Review Assessment: Thoroughness In Paper Reading:**

I read the paper at least twice and used my best judgement in assessing the paper.

---

> ### Author Response · Authors · 2019-11-15
> **Response to Review #2**
>
> We thank the reviewer for the review, positive feedback and appreciation of the real robot experiments. We agree with the reviewer that reproducibility is important. The initial paper submission had an algorithm box in the appendix, which might not have been obvious. We have updated the paper to ensure an explicit mention of the algorithm listing is included in the main text.
>
> If the reviewer would like to see additional details in the listing please let us know, and we will adjust it accordingly.

---

### Decision · Program_Chairs · 2019-12-19

**Decision:**

Accept (Poster)

**Comment:**

The authors present a novel stable RL algorithm for the batch off-policy setting, through the use of a learned prior.  Initially, reviewers had significant concerns about (1) reproducibility, (2) technical details, including the non-negativity of the lagrange multiplier, (3) a lack of separation between performance contributions of ABM and MPO, (4) baseline comparisons.  The authors satisfactorily clarified points (1)-(3) and the simulated baseline comparisons for (4) seem reasonable in light of how long the real robot experiments took, as reported by the authors.  Futhermore, the reviewers all agree on the contribution of the core ideas.  Thus, I recommend this paper for acceptance.